



# Velocity response of Petermann Glacier, northwest Greenland to past and future calving events

Emily A. Hill[1], G. Hilmar Gudmundsson[2], J. Rachel Carr[1], and Chris R. Stokes[3]

[1]School of Geography, Politics, and Sociology, Newcastle University, Newcastle-upon-Tyne, NE1 7RU, UK
[2]Department of Geography and Environmental Sciences, Northumbria University, Newcastle-upon-Tyne, NE1 8ST, UK
[3]Department of Geography, Durham University, Durham, DH1 3LE, UK

**Correspondence:** E.A.Hill (e.hill3@newcastle.ac.uk)

**Abstract.**

Dynamic ice discharge from outlet glaciers across the Greenland ice sheet has increased since the beginning of the 21st century. Calving from floating ice tongues that buttress these outlets can accelerate ice flow and discharge of grounded ice. However, little is known about the dynamic impact of ice tongue loss in Greenland compared to ice shelf collapse in Antarctica.

The rapidly flowing ($\sim$1000 m a$^{-1}$) Petermann Glacier in north-west Greenland has one of the ice sheet's last remaining ice tongues, but it lost $\sim$50-60% ($\sim$40 km in length) of this tongue via two large calving events in 2010 and 2012. The glacier showed a limited velocity response to these calving events, but it is unclear how sensitive it is to future ice tongue loss. Here, we use an ice flow model (Úa) to assess the instantaneous velocity response of Petermann Glacier to past and future calving events. Our results confirm that the glacier was dynamically insensitive to large calving events in 2010 and 2012 (<10% annual

acceleration). We then simulate the future loss of similar sized sections to the 2012 calving event ($\sim$8 km long) of the ice tongue back to the grounding line. We conclude that thin, soft sections of the ice tongue >12 km away from the grounding line, provide little frontal buttressing, and removing them is unlikely to significantly increase ice velocity or discharge. However, once calving removes ice within 12 km of the grounding line, loss of these thicker and stiffer sections of ice tongue could perturb stresses at the grounding line enough to substantially increase inland flow speeds ($\sim$900 m a$^{-1}$), grounded ice discharge, and

Petermann Glacier's contribution to global sea level rise.

## 1 Introduction

Dynamic ice discharge from marine-terminating outlet glaciers is an important component of recent mass loss from the Greenland Ice Sheet (GrIS) (van den Broeke et al., 2016; Enderlin et al., 2014). Since the 1990s, tidewater outlet glaciers in Greenland have been thinning (Pritchard et al., 2009; Krabill et al., 2000), retreating (e.g. Carr et al., 2017; Jensen et al., 2016; Moon

and Joughin, 2008), and accelerating (Joughin et al., 2010; Moon et al., 2012), in response to climate-ocean forcing. Marine-terminating glaciers are influenced by ocean warming (e.g. Holland et al., 2008; Mouginot et al., 2015; Straneo and Heimbach, 2013), increased surface air temperatures (Moon and Joughin, 2008), and reduced sea ice concentration in the fjords (Amundson et al., 2010; Khan et al., 2014; Reeh et al., 2001). However, glacier response to ocean-climate forcing is highly variable between regions and between individual glaciers, due to differences in glacier topography and fjord geometry (e.g. Bunce



et al., 2018; Carr et al., 2013; Porter et al., 2014). Moreover, changes at the terminus of these glaciers (i.e. calving or thinning), can reduce basal and lateral resistance which alters the force balance at the terminus, and causes inland ice flow to accelerate. Indeed, 21st century retreat at two large outlet glaciers in south-east Greenland (Heilheim and Kangerlussuaq) was followed by acceleration and ice surface thinning (Howat et al., 2005, 2007; Nick et al., 2009).

5  Floating ice shelves or tongues that extend out from outlet glacier grounding lines can also control a glacier's response to calving events (Schoof et al., 2017). Floating ice adjacent to the glacier grounding line can buttress inland ice, depending on the amount of shear and lateral resistance provided along the ice shelf margins (Pegler, 2016; Pegler et al., 2013; Haseloff and Sergienko, 2018). Consequently, thinning and retreat of ice shelves can reduce backstress, which can perturb the stresses at the grounding line and propagate increases in driving stress inland causing accelerated ice flow. Understanding how glaciers may

10 respond to ice shelf loss is therefore important for estimating future flow speeds and, ultimately, their increased contributions to grounded ice discharge and global sea level rise. Considerable work has focused on the role of buttressing ice shelves on grounded ice dynamics in Antarctica (e.g. Schoof, 2007; Gudmundsson et al., 2012; Goldberg et al., 2009; Reese et al., 2018), but less work has been done on floating ice tongues in Greenland, where large calving events have recently taken place (Hill et al., 2017; Box and Decker, 2011; Rignot et al., 2001).

15  One of the last remaining ice tongues in Greenland is at Petermann Glacier, northwest Greenland. Petermann Glacier is fast flowing ($\sim$1000 m a$^{-1}$: Figure 1), and drains approximately 6% of the Greenland Ice Sheet by area (Rignot and Kanagaratnam, 2006; Hill et al., 2017). Mass loss is predominantly via high melt rates (10-50 m a$^{-1}$) beneath the ice tongue (Rignot and Steffen, 2008; Wilson et al., 2017), and also occurs via large episodic calving events (Johannessen et al., 2013). Formerly the glacier terminated in a 70 km floating ice tongue, but two well-documented large calving events in 2010 and 2012 removed $\sim$40 km of

20 the tongue (Johannessen et al., 2013; Nick et al., 2012; Falkner et al., 2011; Münchow et al., 2014). Contrary to the behavior of glaciers terminating in floating ice elsewhere, large calving events at Petermann Glacier were noted to be followed by minimal glacier acceleration ($<$100 m a$^{-1}$) (Nick et al., 2012; Münchow et al., 2014), suggesting that calving from the seaward parts of the ice tongue appear to have limited impact on flow upstream of the grounding line.

  Several ice tongues have been lost from neighboring glaciers in northern Greenland since the early 2000s (C. H. Ostenfeld,

25 Zacharaie Isstrøm, Hagen Bræ: Rignot et al. 2001; Hill et al. 2017; Mouginot et al. 2015), and with Arctic air and ocean temperatures predicted to increase in a warming climate (Gregory et al., 2004), the question remains: at what point will Petermann Glacier lose its ice tongue and how might its complete removal impact on ice dynamics? Petermann's tongue has been retreating from the end of the fjord ($\sim$90 km from the present grounding line) since the beginning of the Holocene (Jakobsson et al., 2018) and currently resides at its most retreated position in recent history (Jakobsson et al., 2018; Falkner

30 et al., 2011; Hill et al., 2018). More recently (2016), another large rift formed across the ice tongue, suggesting another large calving event is imminent. As Petermann Glacier is fast flowing and drains a large area of the GrIS (6%), it has the potential to contribute to increased ice discharge and ultimately sea level rise, once it becomes grounded. Here, we attempt to answer the question: at what point do large calving events from the Petermann ice tongue cause substantial acceleration (i.e. $>$ 100 m a$^{-1}$ that propagates inland of the grounding line) and increased ice discharge? To do this we use the community finite-element ice

35 flow model Úa (Gudmundsson et al., 2012) to:



i). Infer the stress conditions beneath the glacier catchment and along the ice tongue walls

ii). Test whether the recent small changes in velocity following calving events in 2010 and 2012 can be replicated

iii). Assess the future response (acceleration and ice discharge) of Petermann Glacier to further calving events, and eventual entire loss of the remainder of the ice tongue (Figure 1)

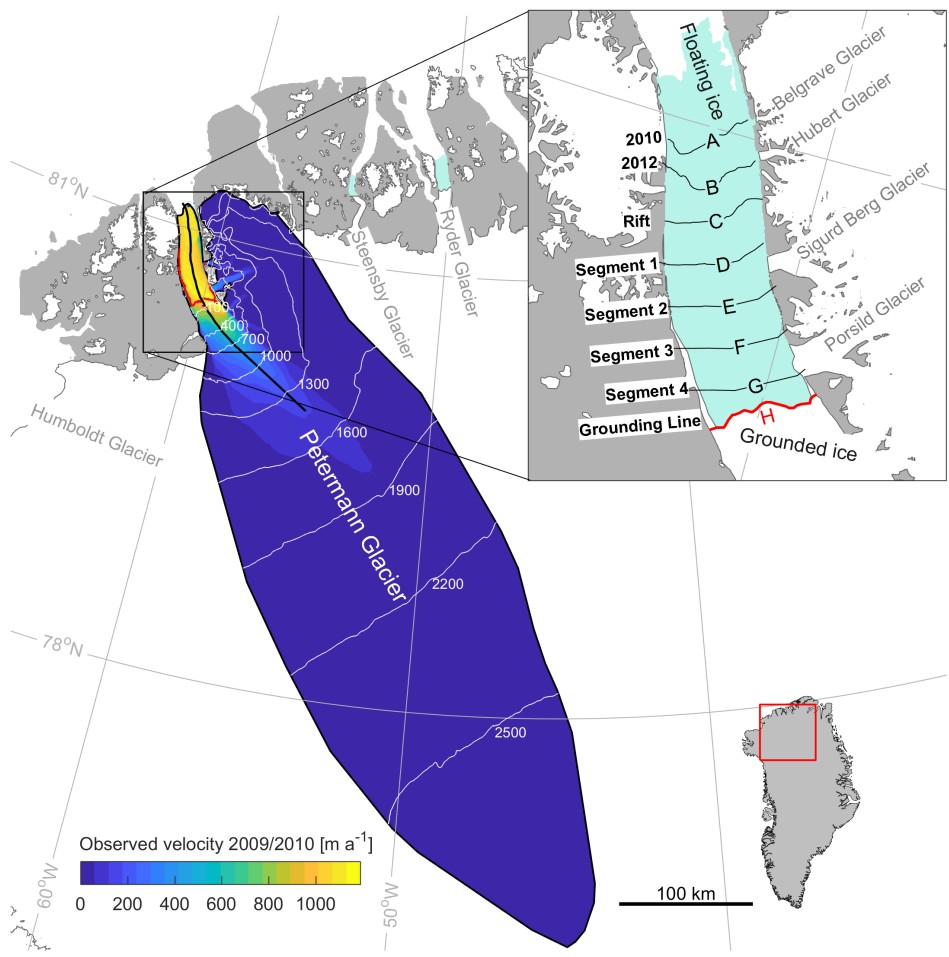

**Figure 1.** Study location, Petermann Glacier, northwest Greenland. Observed ice speeds from the MEaSUREs program (winter 2009/10: Joughin et al. 2010) across the Petermann Glacier catchment, which corresponds with our model domain. White lines show 300 m ice surface contours across the catchment. The thick black line is the glacier centerline and the thick red line is the glacier grounding line. Inset shows the location of newly prescribed terminus positions for each diagnostic perturbation experiment (A-H). Light green shows floating ice, and white is grounded ice.



First, we initialize the model using observational datasets of surface and basal topography. We then invert observed ice velocities prior to the 2010 calving event to determine the initial basal conditions (slipperiness and rheology of the ice). Finally, we perform a series of diagnostic perturbation experiments where we remove sections of the Petermann Glacier ice tongue and assess the instantaneous glacier acceleration and increase in grounding line ice flux.

## 2  Methodology

### 2.1  Data Input

To initialize our model, we use several observational datasets of glacier geometry and ice velocity. Ice surface topography of Petermann Glacier was taken from the Greenland Ice Sheet Mapping Project (GIMP) Digital Elevation Model (DEM) (Howat et al., 2014). We also used this to calculate the surface drainage catchment, using flow routing hydrological analysis in the MATLAB TopoToolbox (Schwanghart and Kuhn, 2010). The defined catchment is ~85,000 km$^2$, extends approximately 550 km inland of the grounding line, and encompasses the tributary glaciers flowing into the east side of the Petermann Glacier ice tongue (Figure 1). Ice thickness and basal topography were taken from the Operation IceBridge BedMachine v3 dataset (Morlighem et al., 2017). These data were generated from satellite derived radar ice thicknesses, ice motion, and the mass conservation method, to resolve the basal topography and ice thickness of the Greenland Ice Sheet (Morlighem et al., 2017). This version also includes high resolution bathymetry of Petermann Glacier fjord seaward of the ice tongue (Mix et al., 2015; Jakobsson et al., 2018; Morlighem et al., 2017). All three topographic datasets have a resolution of 150 m, and a nominal date of 2007, which precedes the large calving event in 2010.

Firstly, to initialize our model via inversion (see Section 2.3), we require annual ice velocities prior our first experiment, which is the calving event in 2010 (Table 1). These were taken from winter 2009/10 from the Greenland MEaSUREs dataset (Table 1: Joughin et al. 2010). This dataset has a resolution of 500 m, and an average error of 8 m a$^{-1}$ across the entire Petermann catchment, which increases to 18 m a$^{-1}$ along the floating ice tongue (Table 1). To validate our modeled velocity changes in response to calving events in both 2010 and 2012, we require observed velocities from the years preceding and succeeding these events. Velocities from winter 2009/10 (used for inversion), also act as our baseline velocities that we compare with observed velocities after each calving event. However, Greenland wide velocities for the winter following the 2010 calving event (2010/11) are not readily available. Instead a series of datasets exist that cover select regions of the ice sheet, derived from feature- or intensity-tracking of optical Landsat imagery (Howat, 2017; Rosenau et al., 2015), or synthetic aperture radar (SAR) imagery (Joughin et al., 2010). We do not use optical Landsat 7 ETM+ derived velocities, because their coverage is restricted to mid-summer, which may reflect seasonal speedups rather than the inter-annual impact of large calving events on ice velocity. Additionally, Landsat derived velocities may have errors associated with cloud cover and or the scanning line correction image banding from May 2003 onwards. Instead, we use a combination of SAR derived velocities (TSX and PALSAR) which are not limited to the summer months, and benefit from higher resolution, and more frequent repeat pass imagery (Table 1). High resolution TSX imagery (100 m) was acquired from the MEaSUREs program (Joughin et al., 2010), but is limited to 11-45 km inland of the grounding line at Petermann Glacier. To supplement this we also used PALSAR derived



**Table 1.** Velocity data sources for Petermann Glacier

| Dataset | Year | Sensor(s) | Resolution (m) | Catchment Error (m) | Use |
|---|---|---|---|---|---|
| MEaSUREs Greenland wide winter velocity NSIDC (Joughin et al., 2010) | 2009/10 | ALOS TerraSAR-X | 500 | 8 m a$^{-1}$ | Model inversion Baseline initial velocities |
| MEaSUREs Greenland Ice Velocity: Selected Selected Glacier Site Velocity Maps from NSIDC (Joughin et al., 2010) | 2010/11 | TerraSAR-X | 100 | 4 m a$^{-1}$ | Validate modeled change post-2010 calving |
| ESA Greenland Ice Sheet CCI project IV Greenland margin winter velocities (Nagler et al., 2016) | 2010/11 | PALSAR | 500 | 16 m a$^{-1}$ | Validate modeled change post-2010 calving |
| MEaSUREs Greenland wide winter velocity NSIDC (Joughin et al., 2010) | 2012/13 | RADARSAT-1 TerraSAR-X TanDEM-X | 500 | 3 m a$^{-1}$ | Validate modeled change post-2012 calving |

velocities (Table 1: Nagler et al. 2016), which provide additional coverage along the western half of the floating ice tongue. Average errors across the catchment are 4 m a$^{-1}$ and 16 m a$^{-1}$ for TSX and PALSAR respectively (Table 1). Greenland wide, winter 2012/13 (post-2012 calving event) velocities were also acquired from the MEAsUREs program (Table 1: Joughin et al. 2010).

To determine observed velocity change, we difference velocity fields after each calving event (2010/11 and 2012/13) from initial baseline velocities (2009/10) (Figure S1). For 2009/10 to 2010/11 speedup along the ice tongue averages 29 m a$^{-1}$, but further inland of the grounding line, noisy and unphysical velocity differences (Figure S1) indicate that there was no coherent velocity change, and it is unlikely velocity changes propagated far inland. For clarity we present centerline velocity profiles in Figure 2, which show that increases in speed were limited to the lower portions of the ice tongue. Observed velocity estimates

presented here after the calving event in 2010, are within the range of previous studies, which showed a 30-125 m a$^{-1}$ speed increase along the ice tongue (Johannessen et al., 2013; Nick et al., 2012; Münchow et al., 2014), and limited change further inland (Nick et al., 2012). After the calving event in 2012, velocity increases averaged 79 m a$^{-1}$ along the ice tongue, and propagated further towards the grounding line (Figure 2 and Figure S2).

## 2.2   Model Initialization

To model the response of Petermann Glacier to ice tongue loss we use the finite-element model Úa (Gudmundsson et al., 2012). Úa solves equations of ice dynamics using the shallow ice-stream approximation (MacAyeal, 1989; Morland, 1987), a Weertman-sliding law, and Glen's flow law. The model has been previously used to understand glacier behavior following ice shelf loss in Antarctica (De Rydt et al., 2015), and ice tongue collapse at the Northeast Greenland Ice Stream (Rathmann et al., 2017). It has also been used to assess the impact of buttressing ice shelves around Antarctica (Reese et al., 2018).



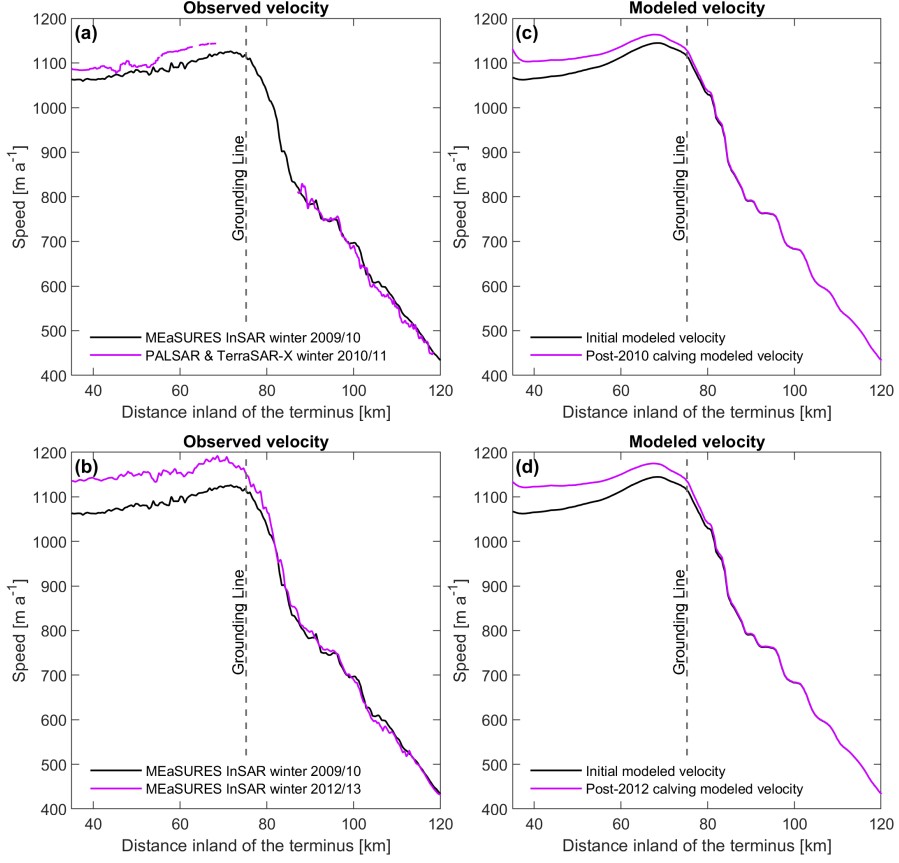

**Figure 2.** Observed and modeled velocities along the Petermann Glacier centerline before and after observed and modeled calving events in 2010 and 2012. **a)** observed initial (2009/10: black line) and post-2010 winter velocities (2010/11: purple line). **b)** observed initial (2009/10: black line) and post-2012 winter velocities (2012/13: purple line). **c)** and **d)** show initial (black line) and post-calving experiment (purple line) model velocities for 2010 and 2012, respectively. Note the small change in both observed and modeled velocities

Previous modeling studies at Petermann Glacier have been conducted using a one-dimensional flowline approach (Nick et al., 2012). Here we aim to expand on earlier work at Petermann to assess if Úa, a two horizontal dimensional, vertically integrated approach, can also replicate the observed velocity response to calving events in 2010 and 2012, and then be used to estimate the impact of future calving events on ice flow and discharge. This model is advantageous over a flowline approach, as it allows

5    us to account for stresses in both horizontal dimensions, which can better assess the impact of ice shelf changes on the force balance at the glacier grounding line (Gudmundsson, 2013).

To set-up the model, we use the surface velocity catchment (Figure 1) as the outer computational boundary, and impose Dirichlet (essential) boundary conditions by fixing velocities to zero inland of the ice divide. Nunataks and rock outcrops along the east side of the ice tongue were digitized using Landsat 8 imagery and treated as holes within the mesh. Ice velocities along

10   the nunataks' boundaries were set to zero, i.e. no-slip boundary condition. Initially, it was less clear what type of boundary





condition to impose along the margins of the ice-shelf where it is in contact with the side walls. We began by imposing a free-slip boundary condition along the ice tongue margins, but conducted further runs with no-slip boundary conditions along the side walls to determine which boundary conditions were most appropriate (see Section 2.4).

The initial calving front boundary was the location of the terminus in 2009, digitized from Landsat 7 ETM+ imagery, and in all cases a Neumann (natural) boundary condition was imposed along the terminus. Using this computational domain, and the finite element mesh generator *Gmsh* (Geuzaine and Remacle, 2009), we generated a high resolution mesh, with 58,000 linear (3-node) elements, and ~30,000 nodes (Figure S3). The unstructured mesh capabilities of Úa allow us to refine the mesh based on the observed velocity field. Where ice speeds are fastest ($> 500$ m a$^{-1}$), primarily along the ice tongue, element sizes are 0.75 km, whereas element sizes inland have a maximum size of 2.7 km. Overall the mean element size is 1.52 km, with a median of 1.4 km. We also increase the mesh resolution of the slower flowing ($<500$ m a$^{-1}$) tributary glaciers to the east of the Petermann Glacier ice tongue to 0.75 km. Topographic datasets (surface, bed, and ice thickness), and pre-calving observed ice velocities (winter 2009/10) were mapped onto this mesh using linear interpolation.

## 2.3 Model Inversion

Before modeling changes in the flow speed of Petermann Glacier due to perturbations in the calving front position, we must first estimate the prior stress conditions. We use Úa to invert the known velocity field (winter 2009/10) before the calving event to estimate the basal slipperiness ($C$) and ice rate factor ($A$) across the catchment. We simultaneously invert for parameters of $C$ and $A$. To estimate these parameters, Úa uses a standard methodology whereby a cost function involving a misfit term and a regularization term is minimized. The gradients of the cost function with respect to $A$ and $C$ are determined in a computationally efficient way using the adjoint method. Here we used Tikhonov regularization involving both amplitude and spatial gradients of $A$ and $C$. Values of regularization parameters were varied by orders of magnitude between 1 and 10,000, and then within range. We also experiment with different sliding law exponent values of $m$ (1,2,3,4,5,7,9) and find the results of our diagnostic experiments to be insensitive to the value of $m$ (Figure S3). We set the stress exponent in Glen's Flow law to $n = 3$.

To begin with, we invert the model using a fixed zero velocity condition along the outer catchment boundary only, and we allow the Petermann Glacier ice tongue to have free-slip boundary conditions (Section 2.2). In the following section we discuss two additional inversion experiments where we vary the boundary conditions along the ice tongue. The model was inverted until the misfit converged which was after 120 iterations. Resultant model velocities ($U_{\mathrm{mod}}$) are in good agreement with observations ($U_{\mathrm{obs}}$) as shown in Figure 3. The mean percentage difference between observed and modeled velocities is 26%, which equates to an absolute difference of 11 m a$^{-1}$ (Table 2). Absolute mean velocity difference increases to 28 m a$^{-1}$ in areas flowing faster than 300 m a$^{-1}$ and to 66 m a$^{-1}$ along the floating ice tongue, which is only 7% of the average ice tongue speed (967 m a$^{-1}$). Flow velocities are not well resolved along the far north-eastern tributary glacier which is due to thin ice thicknesses and poorly resolved bed topography from interpolation (errors of ~150 m).

By inverting the known velocity field (winter 2009/10), we can infer the basal conditions beneath Petermann Glacier. To our knowledge, a catchment scale assessment of the basal slipperiness and ice stiffness has not been previously documented



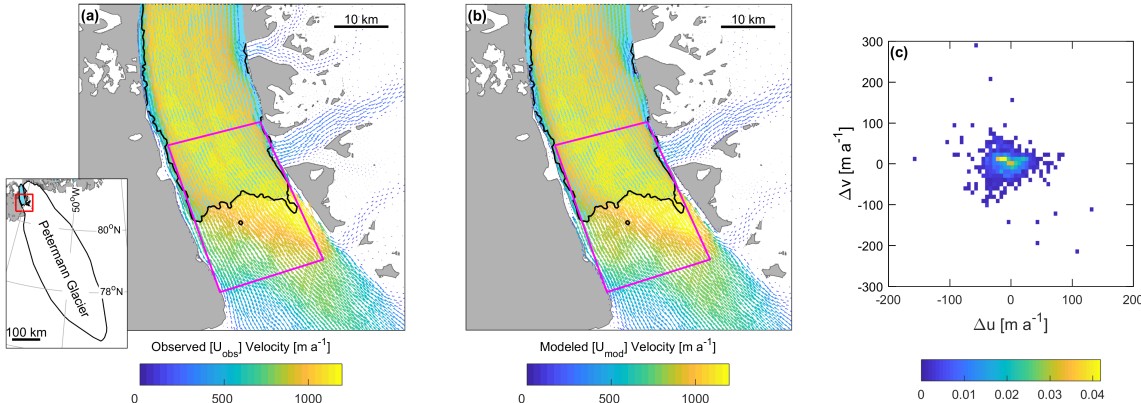

**Figure 3.** Observed ($U_{\mathrm{obs}}$) winter velocities 2009/10 **(a)** in the proximity of the grounding line, **b)** Modeled ($U_{\mathrm{mod}}$) velocities. **(c)** shows a normalized bivariate histogram of the velocity residuals which are the difference between modeled and observed velocities in the vicinity of the grounding line (magenta). $\Delta u = u_{\mathrm{modeled}} - u_{\mathrm{observed}}$ and $\Delta v = v_{\mathrm{modeled}} - v_{\mathrm{observed}}$. $u$ and $v$ are $x$ and $y$ components of the velocity vectors respectively.

for this region. Some studies have examined the basal thermal state (MacGregor et al., 2016; Chu et al., 2018), or provided Greenland wide slipperiness estimates (Lee et al., 2015). However aside from these, little is known on the stress conditions of Petermann Glacier. Here, we provide a new record of the basal conditions beneath Petermann Glacier, which are important for understanding dynamic glacier behavior. For our initial inversion, the distribution of basal slipperiness ($C$), ice rate factor

($A$), and the misfit between observed and modeled velocities are shown in the first line of Figure 4. Basal slipperiness was on average two orders of magnitude greater within 10 km of the grounding line ($C \approx 7.2 \times 10^{-2}$ m a$^{-1}$ kPa$^{-3}$) than the rest of the grounded glacier catchment (mean $C \approx 1.47 \times 10^{-4}$ m a$^{-1}$ kPa$^{-3}$: Figure 4). Grounded ice across the Petermann catchment is on average stiffer ($A \approx 1.2 \times 10^{-8}$ a$^{-1}$ kPa$^{-3}$) than along the ice tongue ($A \approx 7.4 \times 10^{-8}$ a$^{-1}$ kPa$^{-3}$). However, the misfit between observed and modeled velocities is highest along the ice tongue (Figure 4g), which suggests that ice rheology

parameter ($A$) may not reflect the true stress conditions along the ice tongue. The distribution of basal slipperiness and ice rheology parameter ($A$) for our initial inversion are discussed in more detail in the following section.

## 2.4 Boundary Conditions

In an attempt to improve the misfit between observed and modeled velocities, and accurately replicate the lateral resistive stresses along the ice tongue margins, we conducted runs with both no-slip and free-slip boundary conditions along the side

walls. We then tested which produce the best fit to observed velocities. Alongside our additional inversion (Scenario 1), where no velocity condition was imposed along the ice tongue margins, we inverted the model using two further sets of boundary conditions. These are: Scenario 2) fixed velocities along the western margin of the ice tongue to zero (no-slip) and leave the east margin as free-slip, Scenario 3) fixed velocities to zero (no-slip) along both margins of the floating ice tongue. We then base our assessment of these boundary condition scenarios on three criteria: i) the misfit between observed and modeled velocities,



**Table 2.** Misfit between observed and modeled velocity for each boundary condition scenario

| Boundary Condition Scenario | Mean percentage difference (%) | Mean velocity difference (m a$^{-1}$) | Mean velocity difference (>300 m a$^{-1}$) | Mean velocity difference ice tongue (m a$^{-1}$) |
|---|---|---|---|---|
| 1. Natural ice tongue boundary | 26 | 11 | 28 | 66 |
| 2. Fixed west ice tongue margin to zero | 24 | 12 | 34 | 77 |
| 3. Fixed both ice tongue margins to zero | 20 | 9.4 | 25 | 51 |

ii) observations of the confinement and attachment of the ice tongue to the fjord walls in satellite imagery (i.e. heavy rifting or not), iii) the ability of each set of boundary conditions to replicate the observed velocity response following the 2010 calving event. The former two criteria are discussed in this section, and the third criterion is discussed alongside our model experiments in the following section (Section 2.5). As before, we perform each inversion for 120 iterations until the misfit has converged, and use the same values of: $m$, $n$, and regularization parameters for each scenario. The slipperiness ($C$), ice rate factor ($A$) and misfit distributions ($|U_{\mathrm{obs}}| - |U_{\mathrm{mod}}|$) are shown for each boundary condition scenario in Figure 4. Mean misfits between observed and modeled velocities for each scenario are in Table 2.

Slipperiness values show a similar spatial distribution across all boundary condition scenarios, i.e. increasing towards the grounding line and decreasing further inland (Figure 4 a-c), and do not vary substantially within 10 km of the grounding line (range: $3.8-7.2 \times 10^{-2}$ m a$^{-1}$ kPa$^{-3}$) or across the entire grounded catchment (range: $1.47-2.18 \times 10^{-4}$ m a$^{-1}$ kPa$^{-3}$). In contrast, spatial variations in ice stiffness ($A$) are more obvious between each scenario, which corresponds to differences in the misfit distributions (Figure 4). Average $A$ values along the ice tongue vary by three orders of magnitude between scenarios 1 ($A \approx 7.4 \times 10^{-8}$ a$^{-1}$ kPa$^{-3}$) and 3 ($A \approx 1.4 \times 10^{-5}$ a$^{-1}$ kPa$^{-3}$). Scenarios 1 and 2 show stiff ice across the entire ice tongue, which do not reflect the stiff ice tongue center and weaker margins we would expect from lateral reductions in longitudinal strain rates and ice velocity associated with shearing along the fjord walls (Raymond, 1996). In both cases, velocities do not well reproduce observations along the lateral margins and lower portion of the ice tongue (criterion i: Figure 4 g-h).

In contrast, areas of softer ice ($A \approx 4.6 \times 10^{-5}$ a$^{-1}$ kPa$^{-3}$) exist along the lateral margins of the lower portion of the ice tongue (Figure 4f) when we use a no-slip boundary condition along both ice tongue margins during inversion (Scenario 3). In accordance with previous studies (Nick et al., 2012) and our own observations of satellite imagery, this replicates the apparent weak attachment of floating ice to the fjord walls in the lower/eastern parts of the tongue prior to the 2010 calving event (criterion ii). In Scenario 3, overall mean percentage difference between $U_{\mathrm{obs}}$ and $U_{\mathrm{mod}}$ is also improved by 6% and absolute difference reduced by 15 m a$^{-1}$ along the ice tongue (Table 2). We find that by imposing a no-slip boundary condition along both side-walls of the ice tongue (Scenario 3) allows the inversion procedure to automatically resolve the weak margins of the tongue. Based on criteria i and ii, this Scenario (3) therefore provides the most realistic distribution of ice softness along the ice tongue (criterion ii) and the best model fit to observed velocities (criterion i). These experiments have shown the importance



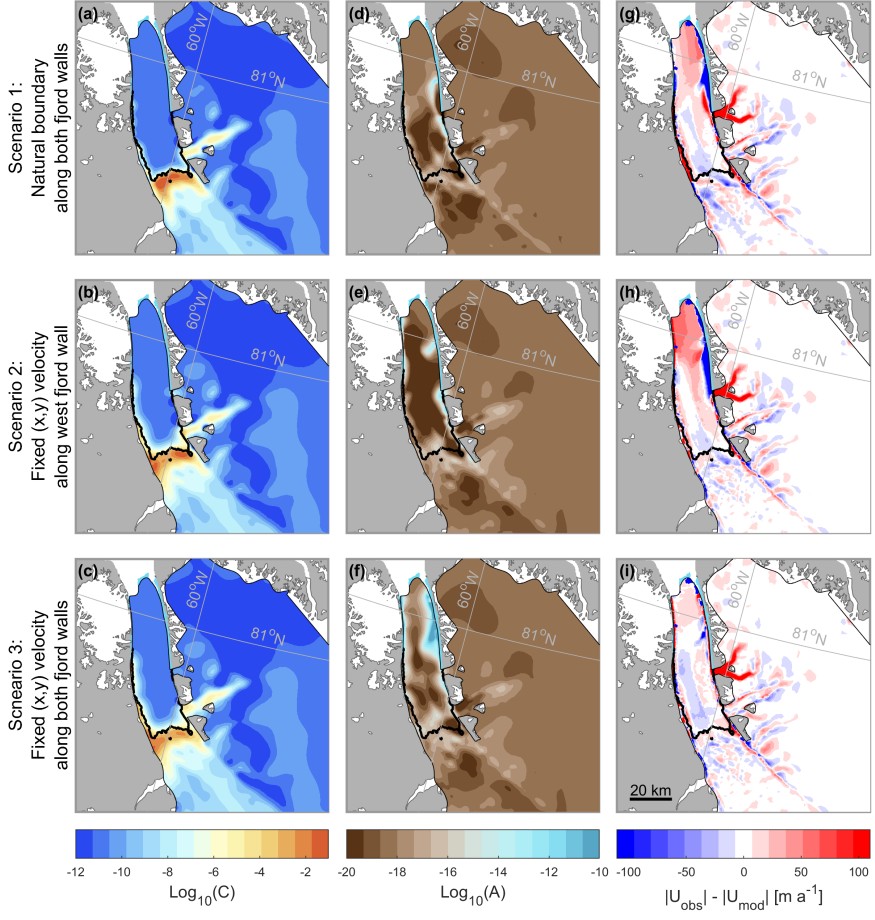

**Figure 4.** Inversion experiments using three sets of boundary conditions along the floating ice tongue. **a-c** show logarithmic calculated basal slipperiness ($C$) for boundary condition Scenarios 1 to 3 respectively, where orange represents highly slippery areas. Glen's flow law rate factor $A$ (**d-f**), where light blue represents soft ice, and brown is stiff ice. The final column (**g-i**) is the absolute difference between observed ($U_{obs}$) and modeled velocities ($U_{mod}$) after inversion using each set of boundary conditions.

of considering boundary conditions, particularly along floating ice margins in this study, for accurately determining lateral resistive stresses and replicating the observed velocity field.

## 2.5 Model Experiments

Following model initialization and inversion, we perform a series of diagnostic experiments (Figure 1) that perturb the calving
5  front position to replicate previous large calving events and potential future loss from the Petermann Glacier ice tongue. We
then examine the instantaneous velocity change with respect to our initial modeled velocities. As the focus of this paper is





the impact of large calving events on glacier velocity, we do not incorporate ice loss via surface and/or basal melting. For each perturbation experiment, we remove all elements from the mesh downstream of the new calving front position, and map all topographic datasets onto the new mesh. We then perform a forward-diagnostic model run, which solves the ice dynamic equations independent of time. In each case, the model is restarted from the previous experiment set up. During all experiments,

grounding line position, boundary conditions, and ice thickness remain fixed.

We start by removing sections of the ice tongue that calved in 2010 and 2012 (Figure 1), for which the new terminus positions were digitized from Landsat ETM+ imagery from 31st August and 21st July respectively. We then assume that the next iceberg to calve from the tongue will follow the path of the rift that formed in 2016. Then, we estimate the glacier response to future calving events, following two assumptions: i) Petermann Glacier will continue to calve episodically, via rift propagation, back

to the grounding line, ii) that future icebergs will be similar sized to previous calving events ($\sim$ 8 km long: Figure 1). Each segment along the ice tongue acts as the new prescribed terminus position, which has a natural boundary condition. In reality, the size and nature of future calving events may vary (e.g. may be a series of small icebergs), but we conduct these experiments to assess the impact of future events similar in magnitude to previous calving. After each diagnostic experiment we calculate the vertically and horizontally integrated flux across the grounding line in Gt, and convert to sea level equivalent (mm) by

dividing by the volume of ice needed to raise global sea levels by 1 mm (361.8 Gt).

For the first diagnostic experiment, we use all three sets of boundary conditions (Scenarios 1-3) proposed in Section 2.4 to fulfill our third criterion (iii) of which scenario best replicates the small increase in velocity observed after the 2010 calving event (Figure 5). Basal slipperiness ($C$), ice rate factor ($A$), and boundary conditions for each scenario were input into this first experiment (2010 calving) and the instantaneous increase in speed presented in Figure 5. We found that the differences

in modeled velocity changes due to calving were relatively small, and the results, hence insensitive to the type of boundary condition applied (see also Section 2.4). This insensitivity to the type of side-wall boundary conditions can be understood to be related to our inverse methodology (see Section 2.3), where $A$ is inferred from measured velocities. In all cases the inversion was able to converge and provide a good fit to observations. Despite this, applying no resistance along the ice tongue margins (Scenario 1: free slip), produces no change in velocity along the tongue, which does not reflect observations (Figure 2 and

Figure S1). However when applying no-slip side-wall boundary conditions, modeled speed increases along the ice tongue (Scenarios 2 and 3) produce a better fit to observed changes i.e. acceleration at the terminus that does not propagate far inland, and average 45 and 37 m a$^{-1}$ respectively. While both Scenarios 2 and 3 appear to adequately reproduce the observed velocity response after the 2010 calving event, we discount Scenario 2 due to the high misfit between modeled and observed velocities (Table 2), and unrealistic ice stiffness along the tongue (Figure 4). Thus, boundary conditions and parameters of

basal slipperiness ($C$) and ice rate factor ($A$) calculated in Scenario 3 are input into our subsequent diagnostic experiments post-2010.



## 3 Results

### 3.1 Response to 2010 and 2012 calving

We have shown that the 2-HD model Úa can reproduce the flow of Petermann Glacier before the large calving event in 2010. Following this we removed sections of the ice tongue, to replicate large calving events in 2010 and 2012, and compare the

5 model results with observed changes in flow speeds (Figure 6a and b).

The iceberg that calved away from Petermann Glacier in 2010 was $\sim$214 km$^2$ and on average 83 m thick (Figure 6). Inverted ice rate factor $A$ reveals that this section of the tongue is softer ($A \approx 2.1 \times 10^{-5}$ a$^{-1}$ kPa$^{-3}$) by three orders of magnitude than the rest of the ice tongue (Figure 7b) or grounded glacier catchment ($A \approx 1.2 \times 10^{-8}$ a$^{-1}$ kPa$^{-3}$). Model results show that removing this section of the tongue was followed by a slight instantaneous increase in speed, ranging from $\sim$65 m a$^{-1}$ (6%

increase of tongue speed) at the terminus, to <20 m a$^{-1}$ between 60 and 80 km along the centreline (Figure 6a). These modeled velocity changes are in very good agreement with observed velocities presented here (Figure 2), and documented in previous studies (Nick et al., 2012; Münchow et al., 2014). After the 2010 calving event increases in speed across the entire ice tongue averaged 29 m a$^{-1}$ and 37 m a$^{-1}$ in observed and modeled velocities respectively (Figure 2). Modeled perturbations in flow speeds did not propagate far inland and averaged only +6 m a$^{-1}$ inland of the grounding line, which is below the average misfit

between observed and modeled velocities (9.4 m a$^{-1}$: Table 2) and therefore indistinguishable from errors. Prior to the calving event, our modeled grounding line flux of 10.12 Gt a$^{-1}$ was within the range of previous estimates by Rignot and Steffen (2008)

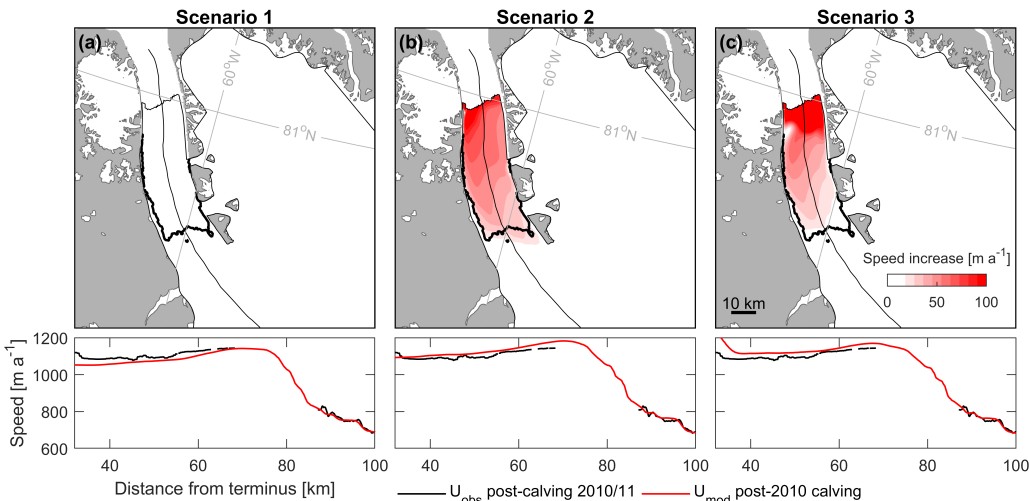

**Figure 5.** Diagnostic perturbation experiments for the 2010 calving event, using three scenarios of boundary conditions applied along the floating ice tongue (**a-c**). Graduated white to red shows speed increase between initial modeled velocities ($U_{\mathrm{initial}}$) and modeled velocities post-2010 calving ($U_{\mathrm{calving}}$). Bottom plots show observed velocities ($U_{\mathrm{obs}}$) post-calving (winter 2010/11), and modeled velocities ($U_{\mathrm{mod}}$) post-2010 calving along the glacier centreline.



(12 ± 1 Gt a$^{-1}$) and Wilson et al. (2017) (10.8 ± 0.52 Gt a$^{-1}$). Limited changes in speed following the 2010 calving event were accompanied by little change in modeled grounding line flux (+0.14 Gt a$^{-1}$) and a negligible increase in sea level rise contribution (Figure 7b).

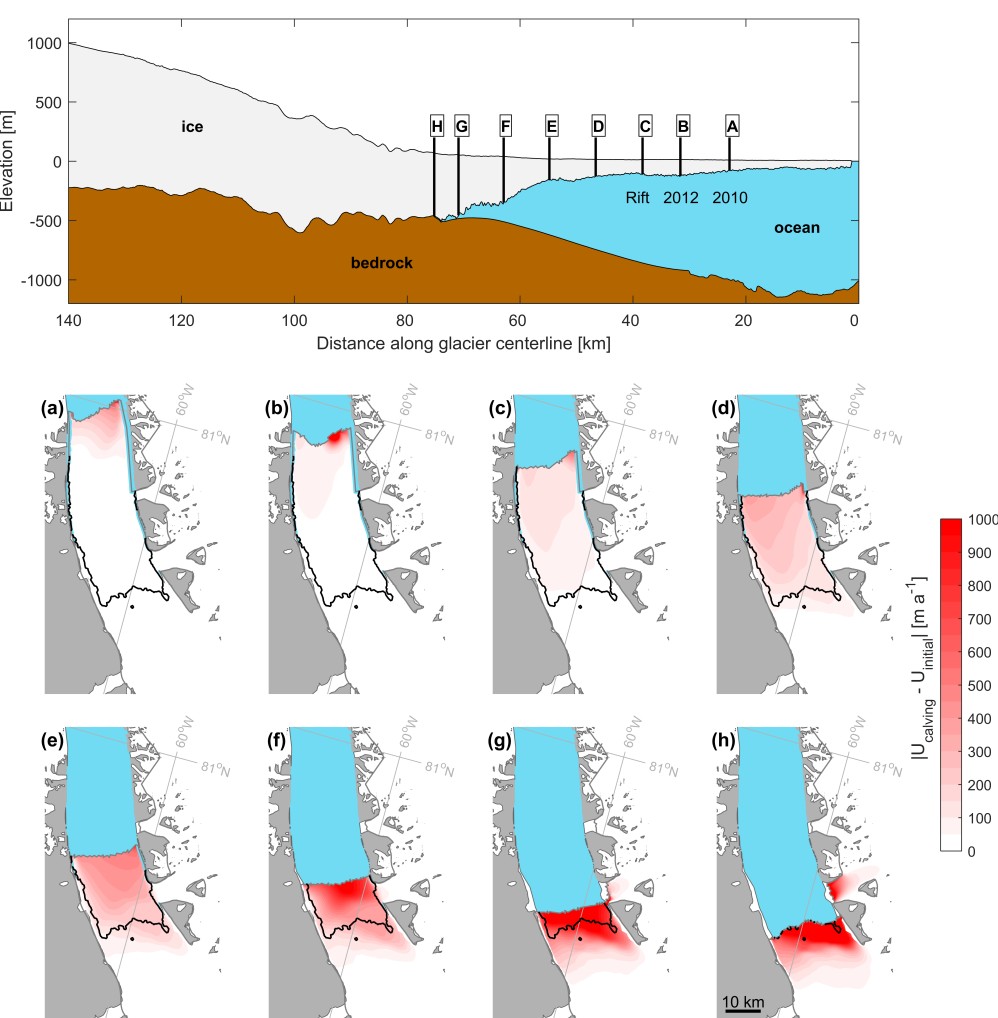

**Figure 6.** Diagnostic perturbation experiments at Petermann Glacier. Top panel shows a cross-sectional centerline profile of Petermann Glacier from the BedMachine v3 dataset (Morlighem et al., 2017). Letters A to H represent sections of the terminus removed for each experiment. A is the 2010 calving event, B is the 2012 calving event, and C is the location of a large rift that formed in 2016. D to G are successive 8 km splices and H is the current grounding line location. Bottom panels **(a-h)** show the modeled instantaneous increase in speed after each experiment with respect to initial pre-calving (before 2010) speeds.





In the next experiment, we removed a 96 km² section of the ice tongue to replicate a subsequent large calving event in July 2012. Importantly this calving event removed a thicker section of the ice tongue that averaged 111 m (Figure 7a). Similar to the modeled dynamic response after 2010, ice flow speeds increased along the ice tongue after the 2012 calving event, and did not propagate far inland of the grounding line (Figure 6b). These modeled velocity changes are consistent with observed

5    velocities in 2012/13 (Figures 2 and S2). Both modeled and observed speeds increased within the range of 3-5% at the terminus, and showed limited change inland of the grounding line (Figure 2). The 2012 calving event was <50% of the 2010 iceberg area, and almost four times softer ($A \approx 9.7 \times 10^{-5}$ a$^{-1}$ kPa$^{-3}$), suggesting it should provide less resistive stress. However, speed increases were 46% greater along the ice tongue than in 2010 (averaging 59 m a$^{-1}$) and propagated further towards the grounding line. Alongside greater acceleration, the 2012 calving event also had a larger impact on grounding line flux, doubling

10   it to 21 Gt a$^{-1}$ and increasing sea level contribution to 0.06 mm (Figure 7b).

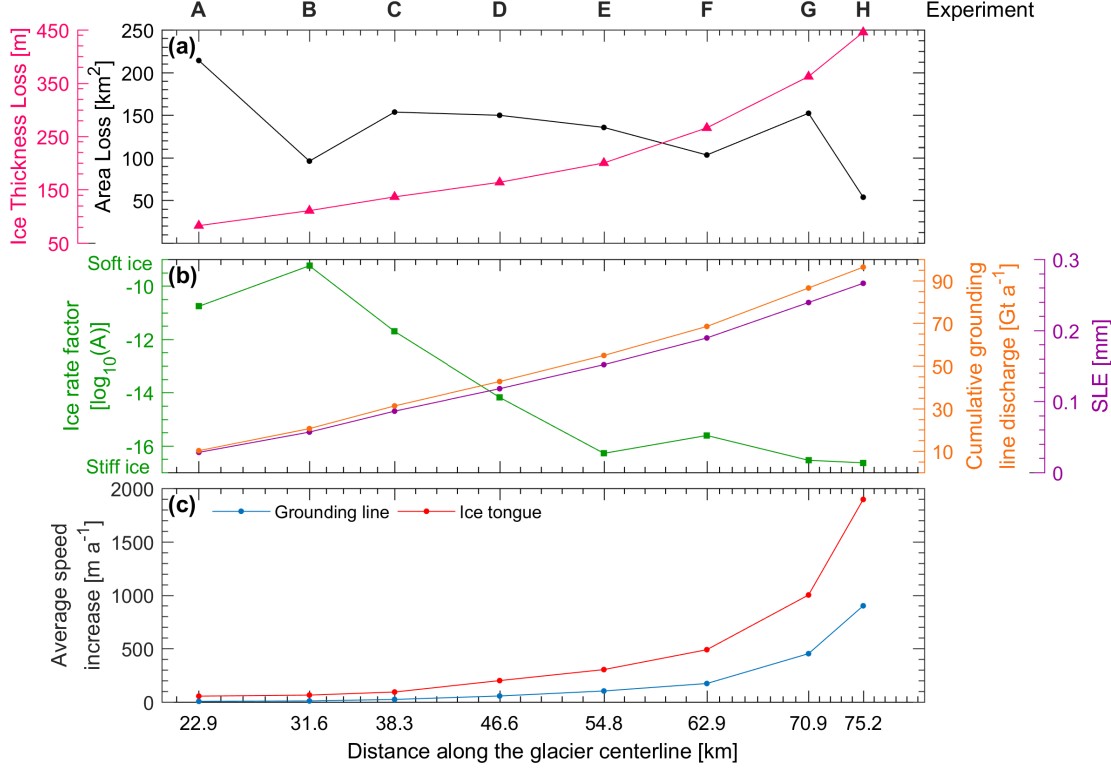

**Figure 7.** Modeled experiment parameters for Petermann Glacier, for each diagnostic experiment (A-H), also shown in Figure 6. **(a)** Black line shows iceberg area lost between each experiment [km²] and magenta is the average ice thickness [m] of each section of ice removed. **(b)** green is the average ice rate factor [$log_{10}A$] across the section of ice removed in each diagnostic experiment. Orange and purple lines represent the cumulative change in grounding line flux and sea level equivalent contribution after the removal of each section of ice. **(c)** Average increases in ice speed across the entire ice tongue (red) and average ice speed within 10 km inland of the grounding line (blue) after each diagnostic experiment.



## 3.2 Response to future calving events

Calving events in 2010 and 2012 had a limited dynamic impact on the ice flow of Petermann Glacier and were followed by <10% acceleration along the ice tongue, and <2% at the grounding line (Figure 2). These modeled findings are consistent with observed velocity change (Figure 2) and previous modeling of the 2010 calving event (Nick et al., 2012). After accurately

replicating the observed velocity response to large calving events in 2010 and 2012, we were confident in the model's ability to predict the future response of Petermann Glacier to further perturbations of the ice tongue.

We conducted six further experiments to analyze the glacier dynamic response (instantaneous change in flow speeds) and grounding line flux, to large calving events. Each of the new calving front positions were approximately 8 km apart along the tongue, and ice loss area averages 125 km (Figure 7a). First, we assume that the next calving event from Petermann Glacier

will fracture along the path of a large rift that formed in 2016, removing a $\sim$154 km$^2$ section of the ice tongue (Experiment C: Figure 6). This segment is also on average 35 m thicker and sturdier ($A \approx 8.3 \times 10^{-6}$ a$^{-1}$ kPa$^{-3}$) than the downstream section of the tongue that collapsed in 2010 and 2012 (Figure 7b). In this case, average increases in speed along the ice tongue were greater (94 m a$^{-1}$) and propagated further ($\sim$30 km from the terminus) towards the grounding line than after previous calving events (Figure 6). Acceleration 10 km inland of the grounding line was double +24 m a$^{-1}$ the acceleration after the 2012 calving

event. Despite this, acceleration inland of the grounding line remained 75% smaller than increases along the ice tongue (Figure 7c), and did not propagate far into the glacier catchment (Figure 6c). After this experiment grounding line flux increased again to 31 Gt a$^{-1}$ and sea level equivalent rose to 0.09 mm (Figure 7b).

Over the subsequent diagnostic calving experiments (D-H), there was a linear increase in average speed change across the ice tongue, as well as a near doubling of average speed increases immediately inland of the grounding line (within 10 km)

between each experiment (Figure 7). After removing the 164 m thick section D from the tongue, average ice tongue speeds increased by 21% compared to initial velocities ($\sim$954 m a$^{-1}$), but increases inland of the grounding line (within 10 km) remained small in comparison ($\sim$+57 m a$^{-1}$). During the following three experiments (E-G), instantaneous average velocity increases across the tongue were more substantial than after previous calving events ranging from 304 m a$^{-1}$ after removing segment E, to increasing by +1000 m a$^{-1}$ ($>$ 100% of initial flow speeds) across the small remaining section of the ice tongue

after Experiment G. Throughout these experiments, higher magnitude increases in speed propagated further into the catchment ($\sim$10-15 km inland of the grounding line) than after previous calving events (Figure 6). Simultaneous to increases along the ice tongue, average speed increases inland of the grounding line (10 km) went from 103 m a$^{-1}$ (Experiment E) to 453 m a$^{-1}$ (Experiment G: Figure 7). Once the last remaining section of the ice tongue was removed (54 km$^2$: H), speed increases were double that of Experiment G, reaching +900 m a$^{-1}$ immediately inland of the grounding line (10 km). Removing the entire ice

tongue, and consequently detaching it from any tributary glaciers also led to a $\sim$530 m a$^{-1}$ speed up at the terminus of Porsild Glacier (Figure 6h).

Alongside linear increases in speed after large calving events, we also note positive trends in the thickness of each calved iceberg, grounding line discharge, and sea level equivalent (Figure 7). Ice thickness along the Petermann Glacier tongue increases from $\sim$50 m towards the terminus to $\sim$500 m at the grounding line (Figure 6: Münchow et al. 2014). In our experiments, the ice



thickness of each segment increased by an average of 221 m (Figure 7a). At the same time, grounding line discharge increased by an average of +10 Gt a$^{-1}$ after each experiment and once the entire ice tongue was removed, cumulative grounding line flux reached 96 Gt a$^{-1}$ and the glacier contribution to sea level rise increased to approximately 0.27 mm per year (Figure 7b). As well as increases in ice thickness along the ice tongue, there is also a general increase in the stiffness of the ice back towards

the grounding line. The ice is generally soft in the lower ~40 km of the tongue (Figure 4f) before ice rate factor ($A$) values decrease by 1-2 orders of magnitude during the last five experiments (E-H: Figure 7b). Importantly grounded ice immediately inland of the grounding line (within 10km) is stiffer than the entire ice tongue.

## 4  Discussion and Conclusions

Here, we expand on previous work and provide new insight into the velocity response of Petermann Glacier to past and
future large calving events, and eventual ice tongue collapse. In contrast to the removal of buttressing ice shelves elsewhere in Greenland (e.g. Jakobshavn Isbræ: Joughin et al. 2008; Zacharaie Isstrøm: Mouginot et al. 2015) and from the Antarctic Peninsula (e.g. Larsen B: De Rydt et al. 2015; Scambos et al. 2004), we show that Petermann Glacier was dynamically insensitive to the removal of ~310 km$^2$ of the ice tongue via calving events in 2010 and 2012 (Figure 6). After both calving events there was a limited increase in speed (< 10% of initial flow speeds: Figure 2), that remained below the ~22-25% seasonal
variability in flow speeds observed between 2006 and 2017 (Nick et al., 2012; Lemos et al., 2018). This insensitivity of ice velocities to large calving events can be explained by weak resistance provided by the lower portion of the ice tongue along its lateral margins (Figure 4f: Nick et al., 2012). From this, we can conclude that the section of the ice tongue that calved away in 2010 and 2012 provided little frontal buttressing on grounded ice.

Given that several floating ice tongues have been lost from neighboring glaciers in northern Greenland (Hill et al., 2018),
and the rapid nature of Petermann Glacier's Holocene retreat from the fjord mouth (Jakobsson et al., 2018), it is possible that the ice tongue will continue to calve episodically, and in the not too distant future collapse entirely. We set out to determine at what point future calving events at Petermann Glacier (similar in magnitude to past calving) will cause substantial acceleration and increased ice discharge. The key conclusion of this work is that future calving events (C-E) from the lower portions of the ice tongue (> 12 km from the grounding line) appear to be passive. We attribute the small modeled velocity response (< 100
m a$^{-1}$ increase at the grounding line) to calving events from this lower portion of the tongue to be due to thinner (<200 m) and an order of magnitude softer ice, which provides limited buttressing on grounded ice. Indeed, if the next calving event takes the path of the 2016 rift formation, it is unlikely to substantially accelerate ice flow (Figure 6c), or increase the glacier contribution to grounded ice discharge and sea level rise (Figure 7). However, we find that removing sections of the ice tongue within 12 km of the grounding line (F-H) has a larger impact on ice flow speeds, increasing them by an average of 900 m a$^{-1}$ (96%) after
the entire ice tongue is removed (Figure 6h). Alongside this, cumulative ice flux across the grounding line increases from 31 Gt a$^{-1}$ (Experiment C) to 96 Gt a$^{-1}$ (H: Figure 7b) and cumulative sea level rise could reach 0.27 mm (for event H). Importantly, within 12 km of the grounding line, the thickness and stiffness of the ice tongue increase dramatically (Figure 7). As such, this part of the ice tongue provides greater lateral resistance along the fjord walls and is therefore more effective at buttressing




grounded ice. Removing these sections of ice is thus likely to alter the resistive stresses at the grounding line enough to cause a greater increase in flow speeds that propagate further inland.

Overall, our findings show that Petermann Glacier has not responded dynamically to previous calving events in 2010 and 2012, and is unlikely to accelerate substantially after imminent future calving events (Figure 6c). However, future large episodic

calving events closer to the grounding line have the potential to perturb the stresses acting on grounded ice, and substantially increase flow speeds and ice discharge (Figure 7). Despite substantial increases in speed forecast after the ice tongue is removed, the question remains as to whether acceleration will be short-lived, and the glacier will re-stabilize at the current grounding line position or retreat inland. Similar to when the glacier was buttressed by an ice shelf at the end of the fjord (Jakobsson et al., 2018), it may be that the current ice tongue has allowed grounding line stability, and its collapse will similarly lead to

unstable grounding line retreat. Indeed, the eastern portion of the current grounding line lies within a deep bedrock canyon (Bamber et al., 2013; Morlighem et al., 2017), which may allow for marine ice sheet instability. However, we cannot discount the possibility that an ice tongue may regrow in the future (Nick et al., 2012). Importantly, future work is needed on the transient evolution of the grounding line to assess its stability under future warming scenarios that also incorporate the effect of basal/surface melt and hydrology on calving front evolution.

*Author contributions.* This work was carried out by E. Hill with input from all authors. E. Hill led the manuscript writing, and all authors contributed towards editing the manuscript.

*Competing interests.* The authors declare that they have no conflict of interest.

*Acknowledgements.* We acknowledge several freely available datasets used in this work. These are: the IceBridge BedMachine v3 (Morlighem et al., 2017) and MEaSUREs InSAR ice velocities (Joughin et al., 2010) both from the National Snow and Ice Data Center (NSIDC), and

PALSAR derived velocities from the ESA Greenland Ice Sheet Climate Change Initiative (Nagler et al., 2015). This work was supported by a Doctoral Studentship award to E. Hill at Newcastle University, UK from the IAPETUS Natural Environment Research Council Doctoral Training Partnership (grant number: NE/L002590/1). E. Hill was also additionally supported by a CASE partnership from the British Antarctic Survey.




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
