# Peer review of "Velocity response of Petermann Glacier, northwest Greenland to past and future calving events"

_The Cryosphere, 2018_

## Referee Comment (RC1) · E. M. Enderlin (Referee) · 16 Oct 2018

Summary: Within the past decade, there have been two very large calving events from the floating ice tongue of Petermann Glacier in NW Greenland. Although the size of the calving events were quite large in terms of aerial extent, they had only a minor influence on ice flow across the grounding line. In this paper, the authors use observations from before and after the 2010 and 2012 calving events to solve for the boundary conditions (both basal and lateral) and viscosity of the glacier in a numerical ice flow model. They then use the observationally-constrained model to simulate the response of the glacier to future calving events of comparable aerial extent. They find that the sensitivity of the glacier to terminus change progressively increases as the terminus retreats towards the grounding line. The increase in sensitivity is due to the

increase in both ice thickness and stiffness towards the grounding line, which lead to a greater reduction in buttressing as the terminus retreats inland.

Comments: The paper is well written and I see no flaws in the methodology or the interpretation of the results. The exploration of lateral boundary conditions is thorough and is clearly very important for the accuracy of the (past and) future simulations.

My only somewhat major comment is in regard to clarity in the presentation of the future calving simulations. The results are presented as the nearly instantaneous response of the glacier to the prescribed calving. I have no problem with this analysis but it is not clear whether the calving events are essentially prescribed as one massive calving event of progressively larger size or if there is a relaxation period in between subsequent events. If they are prescribed one immediately after the other, then the results of the sensitivity tests certainly represent the high end-member response of the glacier to calving because you essentially simulate larger and larger calving events with no time for adjustment of the glacier geometry. I feel like this should be made more clear throughout the text.

I did not find any glaring typos or places where the text required notable revisions. There were a few instances of tense-switching in the methods, so I recommend checking that you consistently use past tense. I will note, however, that the presentation of the discharge across the grounding line in terms of sea level equivalent should be in mm per year (or mm aˆ-1, whatever your preference) since you are talking about a rate and not the cumulative contribution over a set time.

---

## Referee Comment (RC2) · Anonymous Referee #2 · 16 Oct 2018

I am neither glaciologist nor modeler, but physical observations related to force and mass balances in air, oceans, ice constitute my academic bread and butter. My subsequent comments should be read with these biases in mind as I am also unfamiliar with reviewing for Open Access journals such as The Cryosphere.

I thoroughly enjoyed reading this manuscript and endorse it for publication, because its language is concise, elegant, and largely clear of disciplinary jargon. The results are novel, exciting, and will stimulate further observational and modeling work at ice shelves in general and those on Greenland like Petermann or 79N Glaciers. The modeling section-2 is particular clear to me in its detailed and clear discussion of model set-up, initialization, boundary conditions, inversions, limitations, and uncertainties. Nevertheless, an equation or two or three should be given to illuminate both text and

results. Graphics are clear and often free of "chart junk" but the authors can improve their presentation by closer alignment with design principles outlined in

https://www.edwardtufte.com/tufte/books_ei

Fig.-2 in the present manuscript serves as an example for poor design and Fig.-7 as an example for good design.

There are few (minor) quibbles or comments that I have with the manuscript:

(p1, line-23) Khan et al. (2014) is an unfortunate and poor choice to reference the role of sea ice in waters adjacent to glaciers, because the papers merely shows crude (25 km resolution) maps sea ice along with erroneous ocean temperatures (2007 stands out as a particular warm water year off 79 N Glacier in the paper only because no ocean temperatures were collected within 300 km of the coast that year). Furthermore, Khan et al (2014) contains none of the dynamical insights of ice-ocean-glacier interactions such as are provided by Shroyer et al (2017) for Petermann Fjord. Please replace a misleading and poor with a good and relevant reference.

(p2, line-16 and line-31) The authors claim repeatedly that Petermann Glacier's drainage area constitutes 6% of the Greenland Ice Sheet (GIS). This number is used incorrectly by many uncritical glaciologists and oceanographers alike. The mistake, I suppose, originates in Rignot and Kanagaratnam (2006) who in their Table-1 list areas of many glacier drainage basins that add up to 1.2 Mil km^2. This is NOT the total area of the GIS which is closer to 1.7 Mil. km^2 as published by Citterio and Ahlstrom (2013). Using the wrong total area, I get a 6.1% for Petermann, when the correct value is 4.3%. The Geological Survey of Denmark and Greenland distributes the correct digital ice mask. Please correct this common mistake.

(p2, line-22 and subsequent) More complete glacier velocities from time series of GPS measurements on Petermann Glacier are contained in Muenchow et al. (2016) and a year-long time series in Ahlstrom et al. (2013).
(p2, line-31) The rift as a precursor for the next calving can be referenced as Muenchow et al. (2016) who first speculated on the next calving location of Petermann Glacier.

(p6, line-6) Thank you for a wonderful descriptions of different ice flow modeling approaches. May I perhaps ask 2-3 descriptive equations (conversation of mass and momentum as well as stress-law?) that could show me concisely and quickly what kind of dynamics (linear vs. nonlinear, time dependence) are contained. Is "shallow stream" perhaps the same as "shallow water" equations that allow, perhaps, low-frequency wave motions?

(p9, line-15) Could not the central channel at Petermann Glacier with its water-filled river in summer partially explain the discrepancies here? Furthermore, the ice-shelf has substantial small scale topography that is captured in both Operation IceBridge radar and altimeter data and to anyone trying to walk on the floating section. It would be amazing (unlikely), if a model could resolve the details of the velocity field of such small scale divergence and convergences in the ice flow of the floating section of the glacier. What would be the role of basal channels in the model and how are they included or not?

(p20, line-4) The Mix et al reference is neither peer-reviewed nor relevant given several peer-reviewed publications that resulted from this expedition, e.g., Jakobsson et al (2018) or Muenchow et al. (2016).

References: Ahlstrøm, A.P., S.B. Andersen, M.L. Andersen, H. Machguth, F.M. Nick, I. Joughin, C.H. Reijmer, R.S.W. van de Wal, J.P. Merryman Boncori, J.E. Box, and others. 2013. Seasonal velocities of eight major marine-terminating outlet glaciers of the Greenland ice sheet from continuous in situ GPS instruments. Earth Systems Science Data 5:277–287, https://doi.org/10.5194/essd-5-277-2013.

Citterio, M. and A.P. Ahlstrom, 2013: The aerophotogrammatric map of Greenland ice masses, The Cryopshere, 7, 445-449.

Muenchow, A., L. Padman, P. Washam, and K.W. Nicholls, 2016: The ice shelf of Petermann Gletscher, North Greenland, and its connection to the Arctic and Atlantic Oceans, Oceanography, 29 (4), 84-95.

Rignot, E. and P. Kanagaratnam, 2006: Changes in velocity of the Greenland Ice Sheet, Science, 311, 986-990.

Shroyer, E.L., L. Padman, R.M. Samelson, A. Muenchow, and L.A. Stearns, 2017: Seasonal contro of Petermann Gletscher ice-shelf melt by the ocean's response to sea-ice cover in Nares Strait.

---

## Short Comment (SC1) · 14 Nov 2018

Generally I found the paper very well written with a high level of details. As I am currently working on a similar study, I read the discussion paper with great interest. However, I trapped over the sentence given on Page 15 Line 4-6:

> *After accurately replicating the observed velocity response to large calving events in 2010 and 2012, we were confident in the model's ability to predict the future response of Petermann Glacier to further perturbations of the ice tongue.*

I have some concerns with this statement and miss a deeper discussion about the drawback of the model approach (e.g. no geometric adjustment). I think these points should be addressed with more detail:

1) As already pointed out by the Review of E. Enderlin, your approach lacks the adjustment of glacier geometry. This should be deeper discussed with respect to grounding line flux and speed-up. For instance, your pre-2010 grounding line flux of 10 Gt a-1 fits well with Rignot and Steffen (2008). They provide a grounding line flux of 12 Gt a-1 for the pre-2010 calving state. Other than given in your manuscript (Page13 Line1) the Wilson et al. (2017) flux of 10.8 Gt a^-1 refers to the post-2010 calving state (they calculated it as a mean for 2011-2015). So this is not in accordance with the value of 21 Gt a^-1 (Page14 Line10) for your 2010-post calving state. Also, I am wondering how do you get a doubling of grounding line flux from 2010 to 2012 as the ice thickness at the grounding line does not change (due to your model approach) and the speed-up at the grounding line is only minor (Page14 Line5) as PG was dynamically insensitive (Page16 Line 13). Within my model, I only get an increase of the grounding line flux of approx. 1 Gt a^-1 (that is 10%; from 10.78 Gt a^-1 to 11.86 Gt a^-1) if I manually impose a speed-up of 10% to the remotely sensed velocities.

2) The inversion inferred rheology refers to the pre-2010 calving event and aims to mimic stiff and soft zones across the floating tongue. At short timescales it might be valid to keep this field unchanged but on longer timescales (the whole period of your calving events are on decadal timescales, I guess) it is expected that the field changes due to new developing features. The treatment of rheology and impact on results should be at least discussed thoroughly.

Technicalities:

Page4 Line13: Please rewrite, so far we do not have an operating ice thickness radar from satellite for earth observations.

Fig. 2: I am wondering why the velocity profiles for the initial, pre- and pos-2010 (and 2012) events all terminate at the same distance. For instance, after the 2010 event the purple velocity profile should terminate 15 to 20 km earlier than the initially modelled velocities.

---

## Author Comment (AC1) · 23 Nov 2018

**Author response to comments from two reviewers**

**Title:** Velocity response of Petermann Glacier, northwest Greenland to past and future calving events.
**Authors:** Emily A. Hill, G. Hilmar Gudmundsson, J. Rachel Carr, and Chris R. Stokes
**MS number:** tc-2018-162

Dear Editor,

We are very grateful to Ellyn Enderlin and an anonymous reviewer for taking the time to review our manuscript and providing constructive comments. We are also thankful for the insights and comments provided in a short comment by Martin Rückamp. All these comments will greatly improve the content and clarity of our manuscript.

Below we reply to both referee comments and the short comment which are in black text and our responses are in blue. We also include changes made to the manuscript at the end of the document.

Emily Hill and co-authors

**Response to reviewer comment 1: Ellyn Enderlin**

Summary: Within the past decade, there have been two very large calving events from the floating ice tongue of Petermann Glacier in NW Greenland. Although the size of the calving events were quite large in terms of aerial extent, they had only a minor influence on ice flow across the grounding line. In this paper, the authors use observations from before and after the 2010 and 2012 calving events to solve for the boundary conditions (both basal and lateral) and viscosity of the glacier in a numerical ice flow model. They then use the observationally-constrained model to simulate the response of the glacier to future calving events of comparable aerial extent. They find that the sensitivity of the glacier to terminus change progressively increases as the terminus retreats towards the grounding line. The increase in sensitivity is due to the increase in both ice thickness and stiffness towards the grounding, which lead to a greater reduction in buttressing as the terminus retreats inland.

Comments: The paper is well written and I see no flaws in the methodology or the interpretation of the results. The exploration of lateral boundary conditions is thorough and is clearly very important for the accuracy of the (past and) future simulations.

We thank Ellyn for taking the time to review our manuscript and for her careful, largely positive, and constructive feedback.

My only somewhat major comment is in regard to clarity in the presentation of the future calving simulations. The results are presented as the nearly instantaneous response of the glacier to the prescribed calving. I have no problem with this analysis but it is not clear whether the calving events are essentially prescribed as one massive calving event of progressively larger size or if there is a relaxation period in between subsequent events. If they are prescribed one immediately after the other, then the results of the sensitivity tests certainly represent the high end-member response of the glacier to calving because you essentially simulate larger and larger calving events with no time for adjustment of the glacier geometry. I feel like this should be made more clear throughout the text.

Thank you for highlighting this ambiguity in the manuscript. We agree that this could be made clearer to the reader. The model experiments were conducted in an entirely diagnostic time-independent mode, in which we remove each segment of the ice tongue immediately after one another. We agree that they could represent the high-end member response, but we are wary of saying that they are definitely higher than the transient response until such experiments have been undertaken. Also, the purpose of this study is not to estimate the long-term transient impact on ice flow and ice discharge but the instantaneous impact of a change in buttressing at the terminus due to removing large sections of the ice tongue. We agree that in reality there is likely to be a period of relaxation and glacier adjustment before another large calving event takes place. We have added a sentence to section 2.5 where we acknowledge this. We have also added another sentence to the end of section 4 where we say highlight that in this study we have only estimated the instantaneous velocity response, and future transient experiments (that we are also currently working on) are needed to estimate the long-term response of Petermann Glacier to future ice tongue loss.

I did not find any glaring typos or places where the text required notable revisions. There were a few instances of tense-switching in the methods, so I recommend checking that you consistently use past tense.

Thank you for highlighting the switching of tenses in the methods. We have now been through and

made sure in all cases we are consistently using the past tense.

I will note, however, that the presentation of the discharge across the grounding line in terms of sea level equivalent should be in mm per year (or mm aˆ-1, whatever your preference) since you are talking about a rate and not the cumulative contribution over a set time.

Thank you for highlighting this mistake. We have amended sea level equivalent to mm a$^{-1}$ throughout the manuscript, including on the y axis label in Figure 7.

**Response to reviewer comment 2: anonymous reviewer**

I am neither glaciologist nor modeler, but physical observations related to force and mass balances in air, oceans, ice constitute my academic bread and butter. My subsequent comments should be read with these biases in mind as I am also unfamiliar with reviewing for Open Access journals such as The Cryosphere. I thoroughly enjoyed reading this manuscript and endorse it for publication, because its language is concise, elegant, and largely clear of disciplinary jargon. The results are novel, exciting, and will stimulate further observational and modeling work at ice shelves in general and those on Greenland like Petermann or 79N Glaciers.

We thank you for taking the time to review our manuscript, for your insightful comments, and for endorsing it for publication. We too are excited by the potential for further ice shelf work to follow this study across the remaining ice tongues in Greenland.

The modeling section-2 is particular clear to me in its detailed and clear discussion of model set-up, initialization, boundary conditions, inversions, limitations, and uncertainties. Nevertheless, an equation or two or three should be given to illuminate both text and results.

Graphics are clear and often free of "chart junk" but the authors can improve their presentation by closer alignment with design principles outlined in https://www.edwardtufte.com/tufte/books_ei

Fig.-2 in the present manuscript serves as an example for poor design and Fig.-7 as an example for good design.

We have updated the layout (Fig.2) to align with the design of Fig. 7. We have now plotted observed and modeled speeds on-top of one another for each calving event. We have also updated the figure caption accordingly. Following advice from the short comment by Martin Rückamp we now also show the velocity datasets based on the terminus position for each event.

There are few (minor) quibbles or comments that I have with the manuscript:

(p1, line-23) Khan et al. (2014) is an unfortunate and poor choice to reference the role of sea ice in waters adjacent to glaciers, because the papers merely shows crude (25 km resolution) maps sea ice along with erroneous ocean temperatures (2007 stands out as a particular warm water year off 79 N Glacier in the paper only because no ocean temperatures were collected within 300 km of the coast that year). Furthermore, Khan et al (2014) contains none of the dynamical insights of ice-ocean-glacier interactions such as are provided by Shroyer et al (2017) for Petermann Fjord. Please replace a misleading and poor with a good and relevant reference.

Thank you for highlighting this more appropriate reference to us. We have now replaced the Khan et al. (2014) reference with Shroyer et al. (2017).

(p2, line-16 and line-31) The authors claim repeatedly that Petermann Glacier's drainage area constitutes 6% of the Greenland Ice Sheet (GIS). This number is used incorrectly by many uncritical glaciologists and oceanographers alike. The mistake, I suppose, originates in Rignot and Kanagaratnam (2006) who in their Table-1 list areas of many glacier drainage basins that add up to 1.2 Mil km2. This is NOT the total area of the GIS which is closer to 1.7 Mil. km2 as published by Citterio and Ahlstrom (2013). Using the wrong total area, I get a 6.1% for Petermann, when the correct value is 4.3%. The

Geological Survey of Denmark and Greenland distributes the correct digital ice mask. Please correct this common mistake.

We apologize for including this inaccurate percentage drainage area of Petermann Glacier. We have now updated this to 4% in both instances.

(p2, line-22 and subsequent) More complete glacier velocities from time series of GPS measurements on Petermann Glacier are contained in Muenchow et al. (2016) and a year-long time series in Ahlstrom et al. (2013).

We have added these references into the introduction and in subsequent places in the manuscript where we refer to the fit between our model velocities and previous observations.

(p2, line-31) The rift as a precursor for the next calving can be referenced as Muenchow et al. (2016) who first speculated on the next calving location of Petermann Glacier.

We have now included this reference.

(p6, line-6) Thank you for a wonderful descriptions of different ice flow modeling approaches. May I perhaps ask 2-3 descriptive equations (conversation of mass and momentum as well as stress-law?) that could show me concisely and quickly what kind of dynamics (linear vs. nonlinear, time dependence) are contained. Is "shallow stream" perhaps the same as "shallow water" equations that allow, perhaps, low-frequency wave motions?

We appreciate the suggestion to include descriptive equations. In the text we have now included the momentum equation of the shallow ice-stream approximation. We note that these are not identical to the shallow water equations. However, similarly to the shallow-water equations we do assume the flow to be 'shallow' and we also use an integrated from of the mass conservation equitation. On the other hand, our approach includes gradients of horizontal deviatoric stresses that are not included in the shallow water equations.

(p9, line-15) Could not the central channel at Petermann Glacier with its water-filled river in summer partially explain the discrepancies here? Furthermore, the ice-shelf has substantial small scale topography that is captured in both Operation IceBridge radar and altimeter data and to anyone trying to walk on the floating section. It would be amazing (unlikely), if a model could resolve the details of the velocity field of such small scale divergence and convergences in the ice flow of the floating section of the glacier. What would be the role of basal channels in the model and how are they included or not?

While we appreciate that the central channel and small scale topography of the ice shelf could have implications for the distribution of stiff and soft ice across the tongue, we feel that this is beyond the scope of this study, largely because these processes are beyond the resolution of the modelling. These maps of basal slipperiness and ice rheology are inverted from 500m resolution velocities, which would not resolve these small-scale features. Furthermore the element size along the majority of the ice tongue is 0.75 km, which we also assume is too coarse to pick out any of the small scale topographic detail (and therefore stress distribution) associated with the central channel. We also appreciate the importance of the basal channels on the distribution of melt rates beneath the Petermann ice shelf. However, this too is beyond the scope of this study, where we are merely focusing on the buttressing of large portions of the ice tongue. We do not apply any basal melting, nor run any transient time-dependent simulations. We simply assess the buttressing forces of the tongue on the instantaneous response to large iceberg calving events. If we were to conduct transient time-dependent simulations, which we are planning, melt rates across the ice shelf and within the basal channels would need careful consideration.

(p20, line-4) The Mix et al reference is neither peer-reviewed nor relevant given several peer-reviewed publications that resulted from this expedition, e.g., Jakobsson et al (2018) or Muenchow et al. (2016).

This reference has now been removed from the text and the reference list.

We agree that over long time scales that there is likely to be a feedback between changes in geometry and ice rheology. However, here we conduct diagnostic experiments and it is inherent in this approach that there is no evolution of any quantities over time. Thus, there is no need to 'evolve' the rheology with time. We are instead merely estimating the instantaneous impact of a change in buttressing forces at the terminus in response to large calving events. Again, we agree that if this was a transient experiment, in which we allow for a periods of relaxation and geometric adjustment, there may be changes in ice rheology to take into account. We have now added an additional sentence to the methods where we stress that alongside no relaxation or geometric adjustment, our ice stress conditions also remain fixed.

Technicalities:

Page4 Line13: Please rewrite, so far we do not have an operating ice thickness radar from satellite for earth observations.

We have removed 'satellite derived'.

Fig. 2: I am wondering why the velocity profiles for the initial, pre- and pos-2010 (and 2012) events all terminate at the same distance. For instance, after the 2010 event the purple velocity profile should terminate 15 to 20 km earlier than the initially modelled velocities.

Following the advice of anonymous referee 2, we have revised this figure. We have now also now masked the velocity data to demonstrate the different distances at which the glacier terminates after each (2010 and 2012) calving event.

[revised manuscript text omitted]

---

## Author Response (AR2)

**Response to minor corrections from the Editor**

**Title:** Velocity response of Petermann Glacier, northwest Greenland to past and future calving events.
**Authors:** Emily A. Hill, G. Hilmar Gudmundsson, J. Rachel Carr, and Chris R. Stokes
**MS number:** tc-2018-162

Dear Emily & coauthors,

thank you for providing your revision and rebuttal to the comments of the two reviewers. Overall, you did address them adequately so that I can accept your manuscript for publication in TC. However, I would like to emphasize that some parts of your manuscript require some clarifications of the technical kind resp. copyediting, see list below. Please consider and incorporate them in the final revised version of your manuscript.

Regards, Olaf

Dear Olaf,

We are grateful for the acceptance of our manuscript subject to technical corrections. We also thank you for the handling of our manuscript, and again thank the reviewers for their helpful comments. Below we respond to each of the minor revisions, and include a tracked changes manuscript and supplement that highlight changes made.

Best wishes,

Emily & co-authors

Fig. 3 caption:

- you write $u_{modeled}$ and $u_{observed}$. Please use same nomenclature as elsewhere in the MS, i.e. $u_{mod}$ and $u_{obs}$.

We have changed this to $u_{mod}$ and $u_{obs}$.

- change "b)" to "(b)"

Done.

- add comma after vectors

Done.

p9l18: ambiguous use of "3.8 - 7.2 x ..." and "1.47 - 2.18 x". Mathematically different from what you want to express (range), please use brackets before "x" operator for the range terms.

Brackets added around the range terms in both cases.

p12l2: "...relatively small, and the results, hence ..." - unclear what you want to say, rephrase.

Now rephrased to "We found that the differences in modeled velocity changes due to calving, using different sets of boundary conditions, were relatively small. Hence, our results are insensitive to the type of boundary condition applied."

p13l29: I suggest to add the percentage of flux increase in change after the "increasing 0.35 Gt/a" to which this value corresponds on relation to the total flux.

We have added the percentage increase from the initial flux.

p15l8: "line was double +24 m/a the acceleration". Unclear. Is 24 already the double value, or would that be 48? Clarify.

We have clarified and changed it to "Acceleration 10 km inland of the grounding line more than doubled to 24 m a$^{-1}$ compared to acceleration after the 2012 calving event (10 m a$^{-1}$)."

Fig 7 caption: add a statement about the letters A-H at the top of the graph, i.e. what their location on the x-axis stands for.

We hope to have clarified this by now saying "Letters A to H represent the points along the glacier centerline at which sections of the terminus were removed for each experiment."

p16: use use experiments (E-G) and Experiments G etc. Please unify. I suggest to use only experiments without capital E.

Changed to be lower case E in all instances.

Supplement

Fig 1: explain xpsn yspn.

Added.

Fig 2: unresolved reference to other figure: Fig ??. $->$ resolve

Removed.

Fig 3: "value of m" $->$ m in math mode

Done.

[revised manuscript text omitted]